# Total Skin Treatment with Helical Arc Radiotherapy

**DOI:** 10.3390/ijms24054492

**Published:** 2023-02-24

**Authors:** Hsin-Hua Nien, Chen-Hsi Hsieh, Pei-Wei Shueng, Hui-Ju Tien

**Affiliations:** 1Department of Radiation Oncology, Cathay General Hospital, Taipei 10630, Taiwan; 2School of Medicine, College of Medicine, Fu Jen Catholic University, New Taipei City 24205, Taiwan; 3College of Electrical and Computer Engineering, National Yang Ming Chiao Tung University, Hsinchu City 300093, Taiwan; 4Division of Radiation Oncology, Department of Radiology, Far Eastern Memorial Hospital, New Taipei City 220216, Taiwan; 5Institute of Traditional Medicine, School of Medicine, National Yang Ming Chiao Tung University, Taipei 11221, Taiwan; 6School of Medicine, National Yang Ming Chiao Tung University, Taipei 11221, Taiwan; 7Department of Biomedical Imaging and Radiological Sciences, National Yang Ming Chiao Tung University, Taipei 11221, Taiwan

**Keywords:** leukemia, lymphoma, HEARTS, helical tomotherapy, total skin irradiation

## Abstract

For widespread cutaneous lymphoma, such as mycosis fungoides or leukemia cutis, in patients with acute myeloid leukemia (AML) and for chronic myeloproliferative diseases, total skin irradiation is an efficient treatment modality for disease control. Total skin irradiation aims to homogeneously irradiate the skin of the entire body. However, the natural geometric shape and skin folding of the human body pose challenges to treatment. This article introduces treatment techniques and the evolution of total skin irradiation. Articles on total skin irradiation by helical tomotherapy and the advantages of total skin irradiation by helical tomotherapy are reviewed. Differences among each treatment technique and treatment advantages are compared. Adverse treatment effects and clinical care during irradiation and possible dose regimens are mentioned for future prospects of total skin irradiation.

## 1. Introduction

Total skin irradiation is a special radiotherapy technique that is commonly utilized to treat primary cutaneous lymphoma or leukemia cutis. Primary cutaneous lymphoma is defined as non-Hodgkin lymphomas with skin presentation only and no extracutaneous disease evidence at diagnosis [1]. Leukemia cutis occurs in patients with acute myeloid leukemia (AML) and chronic myeloproliferative diseases [2,3]. Numerous classification systems are applied for diagnosis, including the Kiel classification, the European Organization for Research and Treatment of Cancer (EORTC) classification for primary cutaneous lymphomas, the WHO-EORTC classification of primary cutaneous lymphomas, and the WHO classification of hematolymphoid tumors. According to the WHO-EORTC cutaneous lymphoma classification, primary cutaneous lymphoma includes a heterogeneous group of cutaneous T-cell lymphomas (CTCLs) and cutaneous B-cell lymphomas. CTCL is the major group of primary cutaneous lymphomas, accounting for more than 75% of primary cutaneous lymphomas. Among CTCLs, mycosis fungoides (MF) is the dominant variation of cutaneous T-cell lymphoma, followed by Sézary syndrome [1,4].

Mycosis fungoides constitutes 60% of CTCLs. Treatment strategies include direct skin treatment and systemic therapies. Direct skin treatment includes topical medication [5], ultraviolet A, ultraviolet B [6], and total skin electron beam radiotherapy [7]. Systemic therapies recommended for mycosis fungoides include immunotherapy, chemotherapy, steroids, interferon, retinoids, and extracorporeal photochemotherapy [8,9,10,11,12,13,14,15]. The 10-year overall survival and the median survival decrease with increasing diagnostic disease stage [1,16,17,18]. Total skin irradiation has been considered a standard treatment for cutaneous T-cell lymphoma [8,19,20,21,22] and has achieved a high response rate of 70–100% [23,24,25].

Radiotherapy, in the form of local irradiation or total skin treatment, can be utilized for disease control not only for primary cutaneous lymphoma but also for secondary cutaneous involvement lymphoma [26]. Additionally, total skin irradiation has also been reported to control acute myeloid leukemia with disseminated leukemia cutis and acute myelogenous leukemia with extensive cutaneous involvement [27,28]. Total skin irradiation has been utilized as an effective CTCL treatment since the 1950s [29,30,31,32,33,34]. Moreover, total skin irradiation is not only applied for curative intent but is also effective for palliative purposes for symptom improvement [27,35,36,37,38,39,40]. Conventional total skin irradiation is delivered with electron beam radiotherapy. As technology evolves, photon beam radiotherapy can be used for total skin irradiation. A combination of electron radiotherapy and photon irradiation has been used for total skin, thick plaque, and lymph node irradiation in advanced cutaneous T-cell lymphoma [19,20,41].

## 2. Method

All data in this study were collected by searching electronic databases. The PICO structure was used to evaluate the clinical question and guide the data search.

### 2.1. Participants

Participants were patients who received total skin irradiation for cutaneous lymphoma treatment. No age or gender limitation was used for article selection. Both patients who received previous radiation treatment before total skin irradiation and patients who received total skin radiotherapy as first-time radiotherapy were included.

### 2.2. Intervention and Comparison

Studies and case reports evaluating total skin irradiation delivered by helical tomotherapy were included in our literature review. No restriction was placed on previous or concomitant topical or systemic treatment. Studies comparing the clinical outcomes and dosimetry results of total skin irradiation delivered by different methods were included.

### 2.3. Outcome

The evaluated outcomes included clinical disease response, treatment adverse effects during and after radiotherapy, and dosimetry evaluation on both the treatment plan and clinical measurement. The treatment technique, including plan setting, bolus utilization, and beam-on time, was also evaluated for comprehensive evaluation.

### 2.4. Search Strategy

The studies in the review were searched from the following 4 electronic databases: PubMed, Cochrane, EMBASE, and Web of Science. Articles between 1 January 2000 and 30 November 2022 were included in the literature review. The search result is presented as a PRSIMA flow diagram.

### 2.5. Results

A total of 107 studies were included in the initial search of electronic databases. Twenty-eight articles were removed after duplication screening. Thirty-nine articles were filtered by the automation tool. Twenty-one articles were excluded due to improper treatment site and intervention. In total, 19 studies were included in the review (Figure 1).

## 3. Total Skin Electron Beam Therapy

Electron beam radiotherapy has advantages of a high dose at the superficial region with a rapid dose fall off after a few centimeters, which is beneficial for superficial lesion treatment. Total skin electron beam therapy (TSEBT) is the most common technique to deliver radiotherapy to whole-body skin. A high response rate and complete response of mycosis fungoides to TSEBT have been demonstrated [24,25,42,43,44].

The prescribed dose of total skin electron beam irradiation has been investigated [7]. According to the Consensus of the European Organization for Research and Treatment of Cancer, the general prescribed dose of 30 to 36 Gy with 1 Gy per fraction is suggested with an attempted goal, and at least 26 Gy at a 4 mm depth of the trunk is desired [8,45]. For TSEBT, several issues need to be considered in treatment practice.

### 3.1. Considerations in Total Skin Electron Beam Therapy

#### 3.1.1. Limited Field Size

At most facilities, electron beams are generated and delivered through linear accelerators. However, the maximum range of the multileaf collimator of the linear accelerator is 40 × 40 cm, which is not large enough to cover the entire human body surface. Karzmark et al. proposed a large-field superficial electron therapy technique based on a special setting of human–Linac distance and gantry angle to cover the full height irradiation region of a person [29]. Considering the divergent characteristics of radiation beams, extending the source-to-surface distance (SSD) can effectively increase the irradiation field size. To deliver electron irradiation to the entire length of the body, a horizontal beam with patients standing far away from the gantry ranging 3–8 m is suggested by the EORTC consensus [8]. Seven meters of SSD is required for the single horizontal beam setting to treat patients with a height of 2 m or more [8,21]. Considering the limited size of treatment rooms, dual-field irradiation has become the fundamental and practical technique of TSEBT to reduce SSD and increase delivery efficiency [46]. With a dual-field irradiation setting, TSEBT can be performed with an SSD ranging from 2.83 to 5 m for adult treatment [44,47,48,49,50,51,52,53,54,55,56]. For pediatric TSEBT, the SSD can be shortened to 2 m [50,57], and may be further reduced to 1.24 m by adjusting the couch position for patients 1 m in height [58]. Other solutions, such as motorized tables, have been investigated [59,60]. Sufficient space for patient to machine setting, a special gantry angle, and careful junction dose calculation are necessary elements to fulfill the requirement of TSEBT.

#### 3.1.2. Uneven Body Surface Dose Distribution

Electron beam radiotherapy deposits a dose at the superficial region that decreases rapidly a few centimeters below the surface, which is suitable for superficial skin lesions. The most commonly applied electron beam energy for TSEBT is 6 MeV. To distribute an ideal electron beam dose at the surface, a flat target surface with a position perpendicular to the electron beam direction is needed. However, human bodies contain uneven skin surfaces corresponding to body structures, skin folding beneath joints, and circumferential skin regions, which are not ideal flat even surfaces, which renders uniform dose delivery difficult.

In addition to the uneven skin surface, irradiation beams from different directions are required to cover circumferential body skin. The additional field-junction dose increases the complexity of the skin dose distribution. The homogeneity of the dose distribution has been investigated through EBT3 gafchromatic films, thermoluminescent dosimeters, and Monte Carlo calculations [61,62,63,64]. Variation in skin dose inhomogeneity on TSEBT was reported [65,66,67]. Monzari et al. reported that the dose uniformity of TSEBT ranged 100 ± 25% with an accuracy of 6% [62]. Weaver et al. revealed dose variation up to 24% of the focal skin area and large dose deviation up to 40% at the perineum and eyelid [61].

#### 3.1.3. Self-Shielding

To overcome the challenges posed by the circumferential and multifolding nature of the human body skin surface, many techniques with various postures, beam energies, and treatment fields have been investigated [21,68,69,70,71]. The most widely applied techniques for TSEBT are the Stanford six-dual field technique and rotational total skin electron irradiation, which require patients to take on specific postures facing in different directions to maximize the exposure of the skin surface. These special postures allow patients to extend and flatten the skin folds during radiotherapy to ensure coverage for the skin of the entire body [29,33,49,72,73,74,75].

Despite the above technique, self-shielding can still cause underdosage regions, including the undersurface of the breasts, the perineum, the ventral surface of the penis, and the soles of the feet [76]. Consequently, the TSEBT dose could vary from 32% to 124% in specific areas [61]. Additional treatment fields to underdosage regions should be supplemented in TSEBT, and the complexity of the junctional dose calculation should be cautiously considered [8].

## 4. Total Skin Treated by Helical Tomotherapy

Helical tomotherapy (HT) is a rotational intensity-modulated radiotherapy with a unique gantry mechanical design that can deliver highly conformal dose distributions to provide an alternative approach for total body irradiation [77,78,79] or total marrow irradiation [80,81,82]. With special designs, such as virtual bolus, complete block and direction block techniques, HT delivers photon beams with highly conformal dose distribution to convex or concave shape targets while effectively protecting organs at risk (OAR) compared with traditional photon beam radiotherapy. Additionally, the technique allows patients to remain in a comfortable and accurate position with better support during long treatment periods. Several studies have demonstrated that HT is a feasible tool for circular target treatment areas, such as the chest wall and scalp [83,84,85,86,87,88,89]. Accurate dose calculation and delivery of tomotherapy have also been verified [79,83,90]. Therefore, HT has been investigated for use in total skin irradiation, and several techniques have been reported: helical irradiation of the total skin (HITS) [91,92], helical arc radiotherapy of total skin (HEARTS) [93] or total skin helical tomotherapy (TSHT) [94], helical skin radiation therapy (HSRT) [95], and helical intensity modulated radiation therapy (HI) [96].

### 4.1. Clinical Application

Helical tomotherapy (HT) for total skin irradiation has been investigated with phantoms since 2009 [91,97,98,99]. Hsieh et al. applied the first HITS technique with central core complete block (CCCB) in clinical treatment in 2013. To ensure the skin surface dose for HITS, a diving suit was proposed for the whole-body bolus effect, and a complete response was reported [92]. After the report of this successful treatment, the number of investigations and evaluations of HITS gradually increased [28,36,92,93,94,95,96,99,100,101,102,103]. However, given the hematologic adverse effects caused by HITS [92], the HITS technique was revised to develop helical arc radiotherapy of total skin (HEARTS) and avoid toxicity. The distance from the PTV to the central core complete block (CCCB) was modified from 2.5 cm to 2.2 cm. The delivery method was a helical arc with tangential delivery to restrict the photon beams to be obliquely incident to the total skin [93].

Helical tomotherapy to the total skin is not only applied for curative intent but also for palliative therapy [36,103], and most patients receiving this treatment are diagnosed with MF. In addition to MF, HEARTS is also delivered to patients with other diagnoses, such as therapy-refractory cutaneous CD4+ T-cell lymphoma, refractory acute myelogenous leukemia with extensive cutaneous involvement, and primary cutaneous T-cell lymphoma [28,92,93,95].

The clinical prescribed dose varies, including a conventional high-dose level of 26 Gy–36 Gy [92,96], a moderate-dose level of 20 Gy [95,99,101], a low-dose level of 10–14 Gy [28,93,94,95,99,100,101,102,103], and an ultralow dose of 4 Gy [36]. The overall response rate is 100%. Complete response was reported in most cases, as shown in Table 1. Significant improvement of previous lesion-related itching symptoms was also demonstrated [36]. Disease-free duration varied from 2 months to 1.5 years after treatment completion according to the accessible data. Both skin-related and systemic adverse effects were reported. Bone marrow suppression should be carefully evaluated in total skin helical tomotherapy.

### 4.2. Bolus and Skin Surface Dose

The skin-sparing effect of photon beams draws attention to the dose distribution of skin targets. Piotrowski et al. reported an excellent homogenous dose distribution to the surface area for helical tomotherapy, with 90.8–110.2% of the prescribed dose [97]. According to previous experience in total body irradiation, a virtual bolus setting is suggested for targets close to the skin for setup error compensation and the overfluence peak generated by inverse planning avoidance [104,105]. Lin et al. evaluated the dose effects contributed by different thicknesses of hypothetic boluses and various actual bolus thicknesses. The surface dose is increased as the hypothetic bolus increased. With 10 mm of hypothetic bolus, the measurement dose on the phantom surface was 89.5%, 111.4%, 116.9%, and 117.7% of the prescribed dose with 0, 1, 2, and 3 mm of actual bolus, respectively. Hsieh et al. proposed a 3 mm diving suit as a bolus for the entire body and Polyflex II tissue-equivalent material at the ears, fingers, and toes. A hypothetical bolus of 1.0–1.5 cm was set at different regions to prevent overhit in inverse planning. The results revealed good and even 95% to 125% distributed doses in the skin of the entire body [92]. Haraldsson et al. applied a 7 mm neoprene bolus and revealed a significantly higher surface dose (57% compared to the setting without a bolus [99]. Haraldsson’s team also demonstrated that 7 mm neoprene is equivalent to a 3 mm thick water bolus. A slightly soaked neoprene wet suit is equivalent to a 4.2 mm thick water bolus [98]. For the clinical treatment of total skin by HEARTS or other similar techniques, the measured skin surface dose was reported as a maximum underdose of 17.2% for an actual bolus applied and 26% without an actual bolus, as shown in Table 2. Rapid relapse was reported by Schaff et al. (2 months) and Kitaguchi et al. (relapse soon), and both studies delivered radiotherapy by helical tomotherapy without an actual bolus. Although the patient number was limited, the effect of skin surface dose variation on local control warrants further investigation.

### 4.3. Clinical Adverse Effects and Management

Eight studies reported adverse treatment effects, and seven studies provided hematologic examination results [92,93,94,95,96,102,103]. Total skin irradiation is a skin-directed therapy, and treatment adverse effects should theoretically primarily consist of skin toxicity. However, systemic effects are also observed during or after HEARTS or other similar treatment techniques.

#### 4.3.1. Clinical Adverse Effects

The reported skin-directed adverse effects of helical tomotherapy include dermatitis, erythema and epitheliolysis, alopecia, onycholysis, nail changes, paronychia, plantar foot pain, and edema of the fingers and toes. Other adverse effects include grade 1–2 mucositis, xerostomia, fatigue, nausea, fever, watery eyes, and body weight loss. Each symptom was present in a small number of diverse patients. One episode of epistaxis was reported, and the symptom self-resolved 40 min later [94]. Dermatitis, alopecia, and mucositis are the most common skin toxicities. Erythema and epitheliolysis were noted in nonhomogenous dose distribution regions, such as the axillary area, inguinal area, and fingers [96]. Edema of the fingers and toes was only reported by one study [100]. Hair loss usually resolves within 3 months after completion of treatment.

Bone marrow suppression, including anemia, leukopenia, and thrombocytopenia, was present in all seven available hematologic examination results studies. The presentation of leukopenia and thrombocytopenia is more prominent than that of anemia. Grade 3–4 leukopenia and thrombocytopenia were reported in most cases. The nadir of leukopenia and thrombocytopenia usually occurred 1–2 months after the completion of HITS. Each reported individual patient toxicity data point is plotted in Figure 2 and listed in Table 3. Thrombocytopenia tends to persist for longer than leukopenia. Kitaguchi et al. applied HSRT to treat the head and neck, trunk and arms, and leg in 24 patients. Eight patents received three sequential portions of irradiation as total skin radiotherapy. However, one planned HSRT of the head and neck was aborted due to remission of the head and neck lesion during earlier leg irradiation. One patient who received HSRT expired 10 months later due to a graft-versus-host reaction after transplant. According to the study, no cytopenia was noted for head and neck and leg HSRT, and bone marrow suppression symptoms mainly presented in patients who received helical skin radiotherapy at the trunks and arms [95].

#### 4.3.2. Bone Marrow Dose Evaluation

The mean dose delivered to the bone marrow was evaluated. The mean dose in the bone marrow correlates with the total prescribed dose. With the HEARTS technique, the mean dose of each part of the bone marrow at 30 Gy was much lower than that at 30 Gy with the HITS technique [93]. The 30 Gy HEARTS technique provided a lower mean bone marrow dose compared with other HITS techniques using a total prescribed dose exceeding 20 Gy [93,95,96,99]. Low-dose HITS at 10–12 Gy was prescribed as an effective clinical treatment with fewer adverse effects. The mean bone marrow dose of 10–12 Gy HITS ranged from 1.66 to 4.2 [93,94,99,100]. However, grade 4 thrombocytopenia occurred even when the mean bone marrow dose was as low as 1.66 Gy [94].

#### 4.3.3. Management

Bone marrow suppression by HEARTS or other parallel techniques is similar to that in patients who receive total body irradiation (TBI). The possible reasons for hematopoietic syndrome in patients treated by HEARTS or other similar techniques could be that hematopoietic progenitor cells are more radiosensitive than pluripotent stem cells and are easily depleted by irradiation [106,107,108]. Additionally, pluripotent stem cells required approximately 30 days to reconstitute neutrophils and platelets [109]. Therefore, prior to recovery after HEARTS or other similar techniques, the care experience for bone marrow suppression due to TBI and accidental radiation exposure can also be applied to these patients. For patients under bone marrow suppression, supportive and specific care according to each patient’s clinical symptoms are needed. Granulocyte colony-stimulating factor (G-CSF) is critical for neutrophil regeneration, and thrombopoietin is critical for megakaryocyte progenitor cell regeneration [110]. Colony-stimulating factors, including granulocyte macrophage colony stimulating factor, G-CSF, and the pegylated form of G-CSF, can be administered to patients experiencing neutropenia. Cytokine treatment not only mitigates symptoms but also has opportunities to shorten symptom duration [111,112]. Blood transfusion with packed red blood cells and platelets is needed for patients with severe bone marrow suppression. A 25 Gy irradiated leukoreduced cellular production is suggested to prevent transfusion-associated graft-versus-host disease, which may be difficult to distinguish under bone marrow suppression conditions [112,113]. Allogenic/syngeneic stem cell transplantation is a treatment option for patients with persistent bone marrow suppression despite treatments [111]. Amiofostine, an FDA-approved radiation protector, has been primarily demonstrated to prevent radiation-induced mucositis, xerostomia, dysphagia, pulmonary fibrosis, or pneumonitis without altering the tumor treatment effect, which may benefit these patients [114,115,116,117]. Blood transfusion and antibiotics can decrease the mean lethal dose [112]. Other supportive care, including parenteral nutrition, antioxidants, oral glutamine, and yeast-derived 1,3/1,6 glucopolysaccharide, can be applied for maintenance. (Figure 3) The reported recovery times ranged from 2 weeks to 1 year [92,93,96,102,103].

### 4.4. Comparison with Total Skin Electron Beam Radiotherapy

#### 4.4.1. Treatment Field

Due to the radiation output field limitation of the linear accelerator, the SSD range from 3 to 8 m needs to be extended for TSEBT to cover large irradiation fields. The increased SSD significantly reduced the skin dose coverage with depth [64]. TSEBT utilizes a dual-field technique to irradiate the patient’s full height. The gantry angle, beam divergence, and junctional dose need to be carefully evaluated for the dual-field technique. Compared with the electron beam technique, the mechanical design of tomotherapy, which is similar to that of tomography, allows patients to easily move forward for full-body height skin irradiation without distance adjustment. The maximum treatment length in one tomotherapy treatment plan is 135 cm. Patients shorter than 135 cm can receive total skin irradiation in one field. For patients who are taller than 135 cm, two treatment fields can ensure that the skin is irradiated over the entire body length. Although the tomotherapy plan remains subject to a field junction, the radiation dose for whole-body irradiation can be calculated accurately using the planning system.

#### 4.4.2. Accurate Position and Better Position Support

To decrease the self-shielding region during radiotherapy, patients who received TSEBT needed to maintain several fixed standing postures with the extremities away from the trunk for a long period. The postures can only be verified with light field or visual markers [8,118,119]. Conversely, the mandatory daily image-guided system of tomotherapy increases the accuracy of the irradiated position for patients who receive HEARTS or other similar techniques. The daily megavoltage computed tomography (CT) image guide provides opportunities for accurate daily setup evaluation and intrafractional body motion observation. For patients who cannot maintain several fixed standing postures with the extremities away from the trunk for a long duration, HEARTS or other similar techniques allow patients to receive irradiation in a comfortable supine position. This technique also provides support for patients with fixation modalities, such as a full body Vac-Lok cushion and thermoplastic mask [28,103].

#### 4.4.3. Planning System

The planning system is important to evaluate the target and organ dose in radiotherapy. Planning systems that utilize various methods and theories for electron beam radiotherapy have been developing since 1975. Currently, the available electron beam dose calculation mainly relies on the pencil beam algorithm and Monte Carlo algorithm. Electron radiotherapy calculation systems are improving the calculation accuracy by considering the dose energy, field size and field junction at each planning system generation [120]. However, the accuracy of the calculated dose in commercialized electron Monte Carlo algorithm planning systems remains limited [121]. Despite the underdevelopment of planning systems, direct dose measurement through radiochromic EBT film and thermoluminescent dosimetry remain the most common practical methods. However, these dosimeters are usually applied to regions of interest in selected patients and cannot be used for real patient internal organ dose measurements. Other approaches, including optically stimulated luminescent dosimetry, radiophoto luminescent dosimetry, and Cherenkov imaging, are under development and investigation [54,122,123].

In contrast, planning systems for photon radiotherapy dose calculation are mature and provide accurate dose evaluation. In HEARTS or other similar techniques, not only can target coverage be evaluated and revised with an image-based planning system, but the dose of OAR can also be calculated with a computer program [28,92,93,94,99,100,102].

#### 4.4.4. Image-Based Planning for Optimizing Dose Delivery and Personalized Treatment

Helical tomotherapy is an image-based radiotherapy. Irradiation targets are determined via contouring on CT images and can simultaneously spare undesired areas, such as previously irradiated regions. Variation in the surface skin dose of TSEBT was revealed [61,66,67]. According to Elsayad et al., the TSEBT surface dose ranged from 0% to 54% of the prescribed dose at the perineum and bilateral sole. The surface dose of other sites ranged from 46% to 123% of the prescribed dose in the standing position [67]. Supplement electron boost is suggested in TSEBT for regions such as the scalp, sole, and perineum [8]. For HEARTS or other similar techniques, the skin region, including the regions mentioned above, can be contoured as targets, and a planned homogenous dose can be optimized and delivered without extra field boost irradiation.

In addition to the inhomogeneous skin dose distribution, sex and weight were demonstrated to have a significant effect on the dose distribution on TSEBT [67]. However, TSEBT applied a similar approach for all patients, ignoring personal characteristics. For HEARTS or other similar techniques, each treatment plan is calculated based on the unique CT image of each patient, and radiotherapy is delivered according to a personalized plan.

#### 4.4.5. Inverse Planning System for Organ-at-Risk Protection and Dose-Painting Technique

The inverse planning system of tomotherapy can not only deliver a homogeneous target dose but also protect OARs with block techniques. The dose received by the OAR, which cannot be accurately estimated in electron beam radiotherapy, can be evaluated with the tomotherapy planning system. Hsieh et al. proposed HITS for total skin irradiation while sparing previously treated regions according to target volume contouring, and they achieved good target coverage and OAR protection [92]. The same team then reported the advanced HEARTS technique to significantly decrease the OAR dose by adjusting the target volumes and planning parameters, including central core complete blocks [93].

With an image-based inverse planning system, simultaneous boost is also available to increase treatment efficiency for total skin treated by helical tomotherapy. SIB-HEARTS was presented for total skin irradiation with a higher dose to the tumor region. The dose-painting technique allows boost to high-risk regions and decreases in normal organs in the same treatment plan [93]. In addition to the initial planning, an adaptive plan and daily treatment dose recalculation are available through the daily CT image [28].

#### 4.4.6. Beam on Time

Piotrowski et al. noted a longer beam-on time of 21 min for total skin treated by helical tomotherapy than the 7 min needed for rotational TSEBT in the Rando Alderson phantom study [97]. Up to 2 h per treatment session was needed for traditional TSEBT. Three positions per day and two days for the Stanford six-dual fields are needed to complete one fraction of total skin irradiation. With high-dose rate mode TSEBT, Parida et al. were able to reduce each session treatment time to 15 min [124]. With unique adaptation to avoid interruptions between each session, Morris et al. were able to complete all six positions and twelve fields within 30 to 45 min. Thick tumors and shielded regions during irradiation need extra radiation boosts [125]. Although the rotational technique has been shown to reduce treatment time compared with the Stanford technique, additional extra boost fields are required for the rotational technique [46]. The beam-on time of TSHT ranged from 46–92 min according to the available data, as shown in Table 4. An in-room time of 3–3.5 h was recorded in Haraldsson’s studies [99,102]. Haraldsson et al. argued that the treatment frequency per day and extra boost field of TSEBT should also be considered in the treatment time [99]. Considering the operation mechanism of tomotherapy and the clinical reported data, the beam-on time of tomotherapy was revealed to be correlated with target length [99]. Thus, a longer treatment time was needed for taller patients. Wang et al. demonstrated that the beam-on time by tomotherapy for total skin was significantly affected by plan parameters, including field width and modulation factors [126]. The beam-on time decreased while the field width increased and the modulation factors decreased and were not affected by the pitch factor on plan. The distance from blocked structures to the planning target volume (dPTV) is important for normal organ protection. An increasing dPTV leads to an increased OAR dose but decreases the total beam on time of TSHT [126]. Under the precondition of good target coverage and good OAR protection, the parameters mentioned above can be adjusted to optimize the best treatment plan for TSHT patients. The Radixact system with the new treatment planning system for helical tomotherapy reduces the time for gantry rotation from 10 s to 6 s compared to the Hi-ART system and increases the output of 1000 MU (monitor units) per minute compared to 850 MU/min for the old system, so the beam-on-time for total skin might be reduced [127]. The comparison of TSEBT and TSHT is listed in Table 5.

## 5. Future Prospects

### 5.1. Options of Treatment Regimens

Total skin irradiation is an effective treatment for MF. In addition to the skin response, peripheral blood involvement is also reported to be responsive to total skin irradiation [128,129]. However, multiple courses may be needed for recurrent disease according to the natural history of mycosis fungoides. Reirradiation may induce prominent adverse effects due to the cumulative irradiation dose [25,130].

To minimize adverse effects during effective treatment, several groups have investigated possible low-dose regimens for TSEBT [99,131,132,133,134,135]. Harrison et al. revealed comparable overall response rates among groups with dose ranges of 10 to <20 Gy, 20 to <30 Gy, and the standard dose (≥30 Gy) [136]. No significant difference was demonstrated between the group with a radiation dose of ≤12 Gy and the normal radiation treatment group [42]. TSEBT with 12 Gy in eight fractions and the standard dose regimen had a similar overall response rate of 87%, but the former had a lower complete response rate of 18% [131]. A 95% overall response rate with low-dose (10 Gy) total skin electron beam therapy was reported [137]. Jeans et al. integrated low-dose and hypofraction treatment and provided patients with a median dose of 12 Gy in three fractions of total skin irradiation [138]. Aral et al. prescribed 6 Gy with 2 Gy per fraction to the hemibody for palliative treatment, and clinical symptoms improved [139]. A dose of 4 Gy in four fractions was delivered to patients without severe acute side effects. A lower complete response rate and lower progression-free survival but a similar disease progression rate were reported [140,141].

According to TSEBT experience, a moderate-to-low dose may be another approach for HEARTS or other similar techniques to provide an effective but less acutely toxic treatment. Ultralow doses may be suitable for repeated treatment and palliative purposes with limited toxicity. Four Gy in two fractions is also suitable as a palliative and comfortable treatment for elderly patients according to the patient’s condition [36].

### 5.2. Exploring the Risk Factors for Adverse Effects

Bone marrow suppression is the most serious adverse effect of total skin irradiated by helical tomotherapy. Close follow-up of hematologic representation during radiation and after completion of therapy is needed. Initially, treatment and supportive care can assist in recovery. Studies of total skin irradiated by helical tomotherapy reported large variations in many detailed factors, such as delivered total dose, fraction size, bolus utilization, daily treatment frequency, weekly treatment frequency, completion of whole-body treatment in one single fraction or irradiated small portion of body followed by part sequential treatment, and interrupted resting duration. All factors mentioned above may suggest a clinical response and adverse effects. However, large variation exists among the regimens, and the patient number is limited. To further investigate the significance of each parameter, more studies and data collection are needed in the future. The correlation among the above factors, disease response, and clinical adverse effects should be further surveyed. Additionally, total skin irradiation followed by transplantation under specific consideration may provide another method to avoid associated hematopoietic toxicity.

### 5.3. Exploring the Influencing Parameters of Disease Control and Treatment Efficacy

The measured skin surface of total skin treated by helical tomotherapy varies. However, the treatment setting without an actual bolus tends to have a lower measured skin dose according to the literature review. Shorter disease relapse duration was also reported in these studies. The possible factors correlated with the adverse effects mentioned above may also affect clinical disease control. The effects of skin surface dose and treatment setting on disease local control warrant additional studies.

### 5.4. Personalized Treatment Plan

The complete response rate of total skin irradiation has been revealed to be correlated with tumor stage [142,143]. Kim et al. also noted that patient age may influence the complete response rate. Patients younger than 57 years old have significantly better 5-year survival rates than patients older than 57 years old [143]. Adjusting the prescribed dose regimen based on the clinical disease presentation and possible risk factors is a reasonable strategy. SIB combined with total skin irradiation at a dose depending on each patient’s condition is a practical approach for personalized treatment.

## 6. Conclusions

Helical tomotherapy is a suitable approach for total skin irradiation. It provides a comfortable and accurate total skin irradiation experience with personalized dose-painting radiation delivery. However, bone marrow suppression is the major adverse effect of total skin treated by helical tomotherapy. Close surveillance of related hematopoietic indices and active treatment involvement are needed. Further investigations are needed to optimize dose-fractionation schemes and balance high response rates with effects on bone marrow.

## Figures and Tables

**Figure 1 ijms-24-04492-f001:**
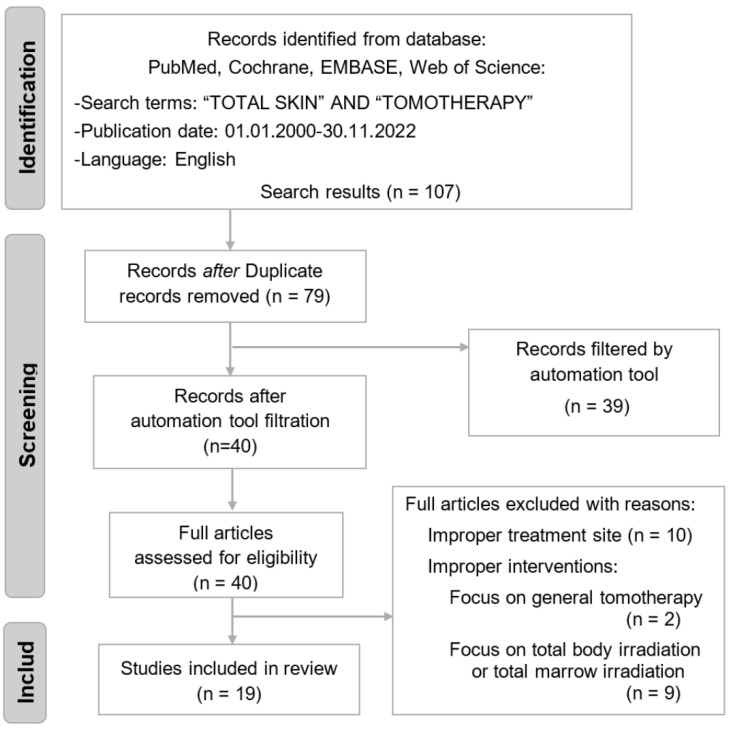
The PRISMA flow diagram for study inclusion.

**Figure 2 ijms-24-04492-f002:**
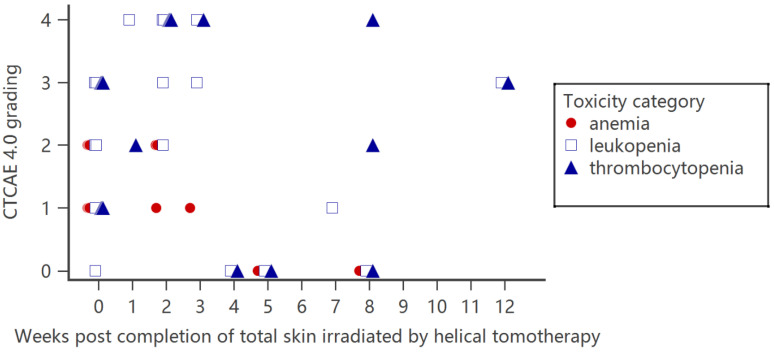
Hematopoietic toxicity severity and presentation time for patients who received total skin irradiation by helical tomotherapy. Each data point represents individual patient toxicity data reported in the articles.

**Figure 3 ijms-24-04492-f003:**
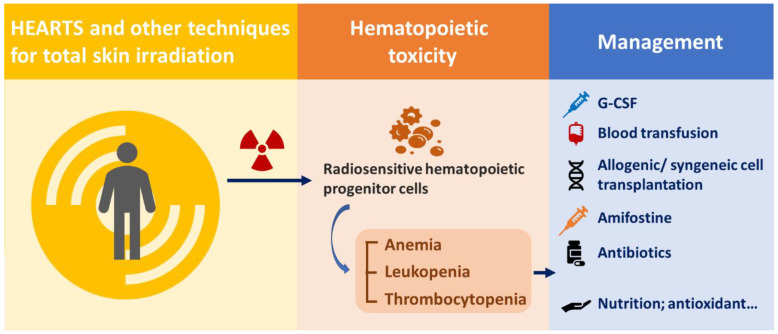
Management of hematopoietic syndrome caused by HEARTS and other techniques for total skin irradiation.

**Table 1 ijms-24-04492-t001:** The reported dose regimens and treatment response of total skin helical tomotherapy.

Study	Patient Number	Total Dose Prescribed	Fractions	Fraction Size	Overall Durations	Treatment Response
Hsieh et al. [92]	1	30 Gy	In 40 Fx with HITS	0.75 Gy	interrupted at 20 fractions, with one week resting, four times per week	CR
Buglione et al. [96]	1	27 Gy to UH body 26 Gy to LH body 22.05 Gy to scalp and eyelids	15 Fx to UH body 13 Fx to LH body 15 Fx to scalp and eyelids	1.8 Gy to UH body 2.0 Gy to LH body 1.47 Gy to scalp and eyelids	5 days a week 23 days split in between	CR
	1	28.8 Gy to UH body 28.8 Gy to LH body	16 Fx to UH body 16 Fx to LH body	1.8 Gy to UH body 1.8 Gy to LH body	5 days a week 15 days split in between	CR
	1	30.4 Gy to UH body 30 Gy to LH body	16 Fx to UH body 15 Fx to LH body	1.9 Gy to UH body 2.0 Gy to LH body	5 days a week 8 days split in between	CR
Haraldsson et al. [99]	1	20 Gy	10 Fx	2.0 Gy	Daily, no reported duration	-
Kitaguchi et al. [95]	6	20 Gy	in 10 Fx	2 Gy	Sequentially treat different parts: Trunk and arms; head and neck; legs no reported frequency or duration	CR: 6
Okuma et al. 2017 [101]	6	10–20 Gy	10 Fx	1.0–2.0 Gy	Over 14 days	-
Hsieh et al. [93]	1	21 Gy to lesions 15 Gy to total skin	15 Fx	SIB-HEARTS 1.4 Gy to lesions 1 Gy to total skin	No reported frequency or duration	CR
Yonekura et al. [103]	1	34 Gy local RT followed by 12 Gy TSHT	17 Fx for local RT 6 Fx for TSHT	2.0 Gy	Over 6 days	CR
Sarfehnia et al. [28]	1	14 Gy TSHT followed by 10 Gy TBI	7 Fx for TSHT 5 Fx for TBI	2.0 Gy	Daily, no reported duration	-
Haraldsson et al. [102]	1	12 Gy	6 Fx	2.0 Gy	Over 30 days	CR
Haraldsson et al. [99]	1	12 Gy	6 Fx	2.0 Gy	Daily, no reported duration	-
Schaff et al. [94]	1	12 Gy	8 Fx	1.5 Gy	4 days per week	PR
	1	12 Gy	6 Fx	2.0 Gy	Daily, no reported duration	PR
Okuma et al. [100]	3	10 Gy	10 Fx	1.0 Gy	Delivered to three parts (trunk, head and neck, legs), irradiate only one part per day no reported frequency or duration	-
Kitaguchi et al. [95]	2	10 Gy	10 Fx	1.0 Gy	Sequentially deliver to three parts: Trunk and arms; head and neck; legs; no reported frequency or duration	CR: 1 PR: 1
De Bari et al. [36]	1	4 Gy	2 Fx	2.0 Gy	No reported frequency or duration	Improved clinical severe itching symptom

Fx: fractions; HITS: Helical irradiation of the total skin; HEARTS: Helical arc radiotherapy of the total skin; SIB: Simultaneous integrated boost; RT: radiotherapy; UH body: upper hemi body; LH body: lower hemi body; TSHT: total skin helical tomotherapy; TBI: total body irradiation; CR: complete response; PR: partial response.

**Table 2 ijms-24-04492-t002:** Skin dose measurement during clinical treatment.

Study	Hypothetic Bolus	Actual Bolus	Measured Equipment	Measured Skin Dose
Sarfehnia et al. 2014 [28]	Not mentioned	No bolus	Gafchromic EBT3 film	Maximum under 25% from TPS
Buglione et al. 2018 [96]	OPTT exist	No bolus	Gafchromic EBT3 film	85–120%
Kitaguchi et al. 2021 [95]	Yes	No bolus	Glass luminescent radiation dosimeter	74–130%
Hsieh et al. 2013 [92]	1.0–1.5 cm	-A 3 mm diving suit -Polyflex II tissue equivalent material: Ears, fingers, toes -Conformal bolus: Trunk lesions	Radiochromic EBT2 film	95–125%
Haraldsson et al. 2018 [102]	Not mentioned	Custom fit, neoprene diving suit of 7 mm thickness	Gafchromic EBT3 film	Median difference from TPS: 4% (SD 11%)
Haraldsson et al. 2019 [99]	8 mm, 0.4 g/cm^3^	-A 7 mm neoprene wetsuit, hood, gloves, and socks of neoprene -A 5 mm water equivalent bolus: Eye lids, forehead	Radiochromic EBT3 film	Mean difference from TPS: Patient 1: 5.3% (SD 11.9%) Patient 2: 1.5% (SD 9.0%)
Hsieh et al. 2019 [93]	1.0–1.5 cm	Diving suit, gloves, socks, head hood	Radiochromic EBT3 film	93–154%

OPTT: PTV portion outside body contour; TPS: treatment planning system.

**Table 3 ijms-24-04492-t003:** Dose regimen, correlated bone/bone marrow dose evaluation, and hematopoietic toxicity for patients treated by helical arc radiotherapy of total skin (HEARTS) or other similar techniques.

Study	Patient Number	Total Dose Prescribed	Mean Dose Evaluation of Bone/Bone Marrow (Gy)	Hematopoietic Toxicity Evaluation Time	Anemia (Grade)	Leukopenia (Grade)	Thrombocytopenia (Grade)
Hsieh et al. [92]	1	30 Gy/40 Fx HITS (0.75 Gy/Fx) interrupted at 20 fractions, with one week resting, 4 times per week	Cervical, thoracic, lumbar spine, sacrum, iliac bone : 5.8, 6.3, 4.0, 4.8, R 8.9/L 8.5	During RT:	1	3	1
				2 ms later:	4	4	4
				The 3rd month after RT:		3	3
Hsieh et al. [93]	Revised plan	30 Gy HEARTS	Cervical, thoracic, lumbar spine, sacrum, iliac bone : 3.6, 3.6, 3.3, 4.0, R 6.1/L 6.2	-	-	-	-
	Revised plan	12 Gy low-dose HEARTS	Cervical, thoracic, lumbar spine, sacrum, iliac bone : 1.5, 1.4, 1.3, 1.6, R 2.4/L 2.5	-	-	-	-
	Revised plan	25 Gy/12 Gy SIB-HEARTS	Cervical, thoracic, lumbar spine, sacrum, iliac bone : 1.9, 1.5, 1.3, 1.4, R 2.1/L 4.0	-	-	-	-
	1	21 and 15 Gy/15 Fx (1.4 and 1 Gy/Fx) SIB-HEARTS	Cervical, thoracic, lumbar spine, sacrum, iliac bone : 2.2, 2.3, 1.9, 3.0, R 3.6/L 3.1	During RT:	1	1	1
				Day 17 post RT:		4	4
				Day 21 post RT:		4 (Nadir)	
				Day 47 post RT:		1	
				Day 60 post RT:			2
Haraldsson et al. [99]	1	12 Gy/6 Fx (2.0 Gy/Fx)	Bone: 4.2	-	-	-	-
	1	20 Gy/10 Fx (2.0 Gy/Fx)	Bone: 7.7	-	-	-	-
Okuma et al. [100]	3	10 Gy/10 Fx (1.0 Gy/Fx)	Bone: 2.27				
Kitaguchi et al. [95]	6	20 Gy/10 Fx (2.0 Gy/Fx) sequentially treat different parts: Trunk and arms; head and neck; legs no reported frequency or duration	Bone in head and neck, trunk and arms, legs group: 12.5, 7.8, 10.6	No mentioned evaluation time	0 (1/6, 16.7%)	0 (0/6, 0%)	0 (0/6, 0%)
					1 (1/6, 16.7%)	1 (0/6, 0%)	1 (2/6, 33.3%)
					2 (2/6, 33.3%)	2 (1/6, 16.7%)	2 (0/6, 0%)
					3 (2/6, 33.3%)	3 (5/6, 83.3%)	3 (2/6, 33.3%)
					4 (0/6, 0%)	4 (0/6, 0%)	4 (2/6, 33.3%)
	2	10 Gy in 10 Fx sequentially treat different parts: Trunk and arms; head and neck; legs no reported frequency or duration	No presented data		0 (0/2, 0%)	0 (0/2, 0%)	0 (0/2, 0%)
					1 (1/2, 50%)	1 (0/2, 0%)	1 (0/2, 0%)
					2 (0/2, 0%)	2 (1/2, 50%)	2 (1/2, 50%)
					3 (1/2, 50%)	3 (1/2, 50%)	3 (1/2, 50%)
					4 (0/2, 0%)	4 (0/2, 0%)	4 (0/2, 0%)
Buglione et al. [96]	1	27 Gy/15 Fx to UH (1.8 Gy/Fx) 26 Gy/13 Fx to LH (2.0 Gy/Fx) 5 days a week 23 days split in between	Bone marrow: 8.5	No mentioned evaluation time	Gr 2 twice during the LH and UH RT; Recovered within 2 ms after RT	2, Recovered within 2 ms after RT	3
	1	28.8 Gy/16 Fx to UH (1.8 Gy/Fx) 28.8 Gy/16 Fx to LH (1.8 Gy/Fx) 5 days a week 15 days split in between	Bone marrow: 10.1	At the end of both UH/LH body RT	1, Recovered within 2 ms after RT	3, Recovered within 2 ms after RT	1
	1	30.4 Gy/16 Fx to UH (1.9 Gy/Fx) 30 Gy/15 Fx to LH (2.0 Gy/Fx) 5 days a week 8 days split in between	Bone marrow: 12.0	At the end of RT	2, Recovered within 2 ms after RT	1, Recovered within 2 ms after RT	3, Prolonged thrombocytopenia, recovered within 6 ms
Schaff et al. [94]	1	12 Gy/8 Fx (2.0 Gy/Fx) 4 days a week	Bone marrow (not including arms): 1.66 (including arms): 2.62	At the end of RT	1		1
				2 weeks after RT	2	2	4
	1	(Local HT) 20 Gy/10 Fx to scalp/buttocks/neck/axilla 10 Gy/5 Fx to back	Bone marrow (not including arms): 2.3 (Including arms): 3.56	2 weeks after RT	1	3	4

Fx: fractions; HITS: Helical irradiation of the total skin; HEARTS: Helical arc radiotherapy of the total skin; SIB Simultaneous integrated boost; RT: radiotherapy; UH body: upper hemi body; LH body: lower hemi body; ms: months.

**Table 4 ijms-24-04492-t004:** Treatment time for total skin irradiated by helical tomotherapy.

Study	Patient	Treatment Part	Duration (Minute)	Average and Total Treatment Time (Minute)
Hsieh et al. [92]	#1	Upper hemi body	48.1 ± 7.9	
		Lower hemi body	8.1 ± 0.8	
Buglion et al. [96]	#1	Upper hemi body 1	22.6 (for 3 Fx)	
		Upper hemi body 2	22.6 (for 12 Fx)	Average: 22.6
		Lower hemi body 1	23.6 (for 10 Fx)	
		Lower hemi body 2	23.6 (for 3 Fx)	Average: 23.6
				Total beam-on time: 46.2
	#2	Upper hemi body 1	27.1 (for 8 Fx)	
		Upper hemi body 2	28.9 (for 8 Fx)	Average: 28
		Lower hemi body 1	40.0 (for 3 Fx)	
		Lower hemi body 2	40.0 (for 13 Fx)	Average: 40
				Total beam-on time: 68
	#3	Upper hemi body 1	27.2 (for 3 Fx)	
		Upper hemi body 2	27.2 (for 13 Fx)	Average: 27.2
		Lower hemi body	29.4 (for 15 Fx)	Average: 29.4
				Total beam-on time: 56.2
Haraldsson et al. [99]	#1 (190 cm)	Wearing suit	Few minutes	
		Total treatment	92	
	#2 (165 cm)	Wearing suit	Few minutes	
		Total treatment	54	Average beam-on time: 73 min for two patients
				in-room time: 3–3.5 h
Yonekura et al. [103]	#1	Upper hemi body	30.8	
		Lower hemi body	15.2	Total beam on-time: 46
Haraldsson et al. [102]	#1 (187 cm)	Upper hemi body	47	
		Lower hemi body	27	Total beam on time: 74 in-room time: 3 h
Kitaguchi et al. [95]	16 patients	Head and neck	12.13 (9.16–21.61)	
	14 patients	Trunk and arms	28.48 (17.92–34.32)	
	13 patients	legs	14.41 (11.9–18.34)	Average total beam-on time: 55.02

#: the serial number of patients.

**Table 5 ijms-24-04492-t005:** Comparison of TSEBT and HEARTS and other similar techniques.

Comparison	TSEBT	HEARTS and Other Similar Techniques
**Radiotherapy**	Electron	Photon
**Treatment position**	Standing, special postures with extremities away from trunk are required	Lie on the couch
**Position confirmation**	Light field and body marks	Image guide assistance
**Planning system**	Most does not rely on planning system	Inverse planning system to ensure dose optimization
**Dose homogeneity**	32% to 124% of prescribed dose; 0 to 54% at self-shielding regions	74–130% of prescribed dose without bolus; 93–154% of prescribed dose with bolus
**Self-shielding regions**	Lower dose irradiation	Ensure dose delivery through targets contouring and image-based planning
**SIB technique**	No	Yes, delivered SIB treatment through targets contouring and inverse planning system
**Personalized treatment**	The complexity of skin dose calculation increases at fields junction and previous irradiation areas	Dose-painting technique is available via targets contouring and inverse planning system
**Hematopoietic toxicity**	Rare	Presented at all available studies
**Beam on time**	Shorter, however, require extra time for extra boost field of TSEBT	Longer, significantly affected by field width and modulation factors

TSEBT: Total skin electron beam therapy; HEARTS: Helical arc radiotherapy of the total skin; SIB Simultaneous integrated boost.

## Data Availability

This is a review article. All the availability data can be accessed at the citated references.

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
