# Peer review of "Total Skin Treatment with Helical Arc Radiotherapy"

_ijms, 2023, doi:10.3390/ijms24054492_

Round 1

Reviewer 1 Report

Dear authors, We have read with great interest your manuscript,  presenting an overview of total skin treatment in helical arc radiotherapy.The article is clear and rich in content. I think it can be published

Author Response

Professor Yamanaka Keiichi                                                          Febunary 19, 2023

Editor-in-Chief,

Dear Professor Keiichi:

On behalf of all authors, I appreciate the time and effort of the editor and reviewers in critiquing our work (ijms-2191646). Attached is the point-by-point response to the reviewers’ comments. We re-submit this manuscript for re-consideration for publication in International Journal of Molecular Sciences. Thank you for your great effort on our work.

Yours sincerely,

Chen-Hsi Hsieh, M.D., Ph.D.

Division of Radiation Oncology, Department of Radiology,

Far Eastern Memorial Hospital,

No.21, Sec. 2, Nanya S. Rd., Banciao Dist., New Taipei City 220, Taiwan

Fax: 886-2- 8966-0906

Phone: 886-2-8966-7000 ext. 1033

Major compulsory revisions:

Reviewer 1

Dear authors, we have read with great interest your manuscript, presenting an overview of total skin treatment in helical arc radiotherapy. The article is clear and rich in content. I think it can be published

Response: Thank you for your time and appreciate for your comments.

Reviewer 2 Report

This is a very interesting paper and you have reviewed the topic comprehensively. There are a few areas where a little more information is required.

The language is difficult to follow in some places, so the meaning is not always clear, for example: “front area” line 63 and “dual filed” line 84.

Introduction: line 44, you need to add immunotherapy to the list of systemic treatments.

TSEBT: field size. You suggest 3-8 metres may be necessary for extended SSD (line 83 and again in line 277). However, how many treatment rooms are that large? Some comment is required about this, and about what most centres who offer this actually do.

TSEBT: electron beam energy. Some statement about which electron energies are preferred as this affects tissue penetration (line 91)

Helical tomotherapy: you comment on different dose used (lines 155-162) but this is not complete without stating the number of fractions and overall duration of treatment. In this section, the presentation of complete response rates might be better in a Table, perhaps with TSEBT as well.

Figure 1: the legend should explain what the individual data points are. Are these individual trials? The legend says “…and similar techniques” – how similar? Perhaps these need listing or referencing here.

Table 2: suggest leukopenia instead of leukocytopenia, in the header row. This table is rather huge and in places difficult to follow, particularly the alignment within each row. However, it does summarise the data well.

3.4.6: beam-on time. Though discussing the principles, this discussion does not give any times at all. How many minutes does it actually take per fraction? How many minutes of beam-on time and how many minutes for set-up time, ideally for both HT and TSEBT.

4.1 treatment regimens: This is quite a crucial section as the higher the prescribed dose, (in general) the higher the bone marrow dose. Your discussion of prescribed doses needs to say more about number of fractions  and overall time. In particular, whether treatment with smaller doses per fraction, or longer overall treatment times might cause less marrow suppression. Perhaps a Table would be good to summarise this data.

4.2: risk factors. You list many factors here, but do not indicate which are adverse and which favourable, and which are the more significant factors. Likewise for age (line 414), is older good or bad?

Conclusion: you have summarised the general situation in terms of alternative treatments well, but I think more emphasis needs to be given to the question of bone marrow dose and where this impacts haematological function, whether this might impact future treatments patients may require. For example, would chemotherapy or other treatments be more difficult to deliver if required subsequently? Ideally, a recommendation about which dose-fractionation schemes are preferred as a good balance between high response rates and effects on bone marrow would be helpful.

Reviewer 3 Report

This paper is a literature review regarding total skin treatment with the helical arc technique with a comparison to total skin electron beam therapy. The language is adequate, however, a few grammar mistakes have been spotted. In general, the manuscript is coherent and well-structured.

Prior to publication, I would suggest a minor revision regarding the following:

·       As mentioned above,  the manuscript should be revised overall for a grammar check by the authors. For instance, the following are mentioned:

-Page 3, line 99: “The homogeneity” not “homogenous”

-Page 3, line 125: Replace “can make photon be delivered” with “delivers photon beams”

-Page 3, line 132: Replace “studies” with “studied”

-Page 4, line 148: Replace “incidence” with “incident”

-Page 4, line 183: Add “without actual bolus as shown in Table 1”

-Page 5, line 199: Replace “epitaxis” with “epistaxis”

-Figure 1, legend: Replace “thromcytopenia” with “thrombocytopenia”

-Page 10, line 305: Replace “Monte Carol” with “Monte Carlo”

-Page 11, lines 336 and 339: Remove “well” before “organ at risk protection”

·       The addition of a PRISMA diagram (Preferred Reporting Items for Systematic Reviews and Meta-Analyses) would be helpful to improve the reporting of the review since the authors don’t mention the literature search methodology.

·       The format of Table 2 should be revised. The number of patients should be mentioned for all the studies included in the Table, whether it was one patient or more, and the number of them that had toxicities. Additionally, in the study of Hsieh et al (ref. 79) it is not clear to  which prescription the toxicities correspond.

·       There is a relevant study regarding the treatment of mycosis fungoides with TSEB technique that the skin dose distribution resulted in complete response. Consider mentioning it: “Diamantopoulos S, Platoni K, Kouloulias V, et al. First treatment of mycosis fungoides by total skin electron beam (TSEB) therapy in Greece. Rep Pract Oncol Radiother 2014;19:114–9”
